# Machine Learning Based Localization in Large-Scale Wireless Sensor Networks

**DOI:** 10.3390/s18124179

**Published:** 2018-11-28

**Authors:** Ghulam Bhatti

**Affiliations:** Department of Computer Science, Taif University, Taif 21974, Saudi Arabia; gbhatti@tu.edu.sa; Tel.: +966-55-748-7635

**Keywords:** wireless sensor networks, localization, random vs. grid placement, simulatoins, Internet of Things (IoT), machine learning algorithms, model fitting, support vector machines, regression

## Abstract

The rapid proliferation of wireless sensor networks over the past few years has posed some serious technical challenges to researchers. The primary function of a multi-hop wireless sensor network (WSN) is to collect and forward sensor data towards the destination node. However, for many applications, the knowledge of the location of sensor nodes is crucial for meaningful interpretation of the sensor data. Localization refers to the process of estimating the location of sensor nodes in a WSN. Self-localization is required in large wireless sensor networks where these nodes cannot be manually positioned. Traditional methods iteratively localize these nodes by using triangulation. However, the inherent instability in wireless signals introduces an error, however minute it might be, in the estimated position of the target node. This results in the embedded error propagating and magnifying rapidly. Machine learning based localizing algorithms for large wireless sensor networks do not function in an iterative manner. In this paper, we investigate the suitability of some of these algorithms while exploring different trade-offs. Specifically, we first formulate a novel way of defining multiple feature vectors for mapping the localizing problem onto different machine learning models. As opposed to treating the localization as a classification problem, as done in the most of the reported work, we treat it as a regression problem. We have studied the impact of varying network parameters, such as network size, anchor population, transmitted signal power, and wireless channel quality, on the localizing accuracy of these models. We have also studied the impact of deploying the anchor nodes in a grid rather than placing these nodes randomly in the deployment area. Our results have revealed interesting insights while using the multivariate regression model and support vector machine (SVM) regression model with radial basis function (RBF) kernel.

## 1. Introduction

Wireless sensor networks are a vital part of the networked world of the future, also popularly known as the Internet of Things (IoT), which will be widely used for pervasive connectivity, control, security, and general awareness of things, big or small, near or remote, and wired or wireless. The deployment of wireless sensor networks in many scenarios and applications require the sensor nodes to be aware of their physical location in order to correlate the sensor data and location. Such a fusion allows meaningful interpretation of sensor data and better utilization of network resources.

Determining the physical location of wireless sensor nodes offers significant challenges due to a number of reasons. The sensor nodes generally have very scarce computational and networking resources. In most, if not all, deployments, the wireless sensor nodes operate on battery power. Since replacing the drained batteries in many large deployments is an infeasible option, a sensor node’s useful life is as long as its battery’s life. In order to reduce the cost per node and minimize the power consumption, a computationally efficient and physically inexpensive (i.e., without requiring special hardware such as GPS card, clock synchronization mechanism, or ranging equipment, etc.) localizing algorithm is required.

The localization of sensor nodes in a deployment requires a small number of these nodes to be initially aware of their own physical location through the use of GPS or simply by manually programmed. These nodes are called anchor nodes. The non-anchor nodes in the network are then localized by using the anchor nodes with an appropriate localization algorithm.

Several localization algorithms have been reported in literature for both indoor and outdoor applications. However, most of the reported work aims at enhancing the accuracy in localizing a single node, static or mobile, in a wireless sensor network (WSN). However, in many applications of these networks, such as routing, cooperative sensing, and service delivery, it is desirable to localize all sensor nodes in order to fully utilize the sensor data originating from those nodes. Due to inherent instability in wireless communications, the ranging based approaches, which aim at estimating the pair-wise distances between the wireless nodes, face a serious challenge in terms of accuracy. The ranging inaccuracy thus introduced propagates and magnifies during the network-wide localization of sensor nodes in an iterative manner.

We have used machine learning algorithms for network-wide localization in large wireless sensor networks. These algorithms do not localize sensor nodes in an iterative manner and so no error propagation takes place during the localization process. In this paper, we define a set of novel feature vectors for localization in WSN, map these vectors onto different regression models, and then use simulations for the performance analysis of these models. We also study the impact of placing the anchor nodes in a grid versus at random locations in the deployment area. We also describe different trade-offs that this approach offers. Finally, we present our results from simulations and discuss the suitability of this approach for different application scenarios.

The remaining paper is organized as follows. We first provide a brief literature review for the reported work related to localization in wireless sensor networks in the next section. Section 3 presents a brief overview of suitability of machine learning algorithms for localization, our rationale for formulating multiple feature vectors to be used with machine learning algorithms, and different factors needed to be considered while using these algorithms. The system models and proposed algorithms are described in Section 4. The performance analysis of the proposed algorithms and subsequent discussion are presented in Section 5. Finally, we conclude in Section 7.

## 2. Related Work

A vast body of work on localization in wireless sensor networks has been reported in the literature. As a result, several approaches and corresponding algorithms have been proposed for localizing wireless nodes by using their communications with neighboring nodes [1,2]. These algorithms can broadly be categorized into two classes, i.e., range based [3] and range-free [4] localization algorithms.

The range-based localization approach aims at estimating the distance between the target node and its neighboring anchor nodes and then applying different algorithms to calculate its location. The distance between the wireless nodes can be measured through a variety of metrics. For example, distance measurements with good resolution can be estimated by measuring the time of flight (that is, the propagation time) of the radio signals while traveling from the transmitting node to the receiver node. That, in fact, is one of the two widely used approaches to estimate the distance among neighboring nodes. Several algorithms exist to estimate the time of flight such as time of arrival (TOA) method [5,6,7,8,9], time difference of arrival (TDOA) method [8,10], etc. Another signal metric, i.e., angle of arrival (AOA), is based on the angle at which a signal from the anchor arrives at the target node. It requires the use of an array of antennas by recording the signal’s TOA at each antenna. The above mentioned methods, however, require specialized hardware, for example, to provide clock synchronization or ultra wide band (UWB) radios for capturing better signal-arrival time in the wireless sensor nodes in order to achieve a reasonable localization accuracy [11,12].

Yet in another approach, the received signal strength (RSS) is used to estimate the distance. It is a useful metric that uses the strength of the received signal to estimate the propagation distance between nodes [7,8,9]. Measuring the RSS is less expensive because this metric can be measured during data communications using low-complexity circuits. In fact, most modern wireless sensor nodes can provide RSS mapping, called received signal strength indicator (RSSI), at no additional cost. Although widely implemented, RSS based method has limited accuracy due to the difficulty in precisely modeling the relationship between the RSS and the propagation distance [9,13]. These RSSI-based localization methods can be classified as those based on fingerprinting [14,15] and those using signal propagation modeling. The latter aspect requires better propagation model that captures the interference and fading caused by multi-path propagation and shadowing in a specific deployment area [16].

Once these distances have been determined, the triangulation technique can then be used to estimate the physical position of the target node. It is worth mentioning that every target node must have at least three anchors as its one-hop neighbors for it to be localized in two dimensions (i.e., on a plane). The accuracy of the estimated location, however, depends on how accurately the distances between the target sensor node and its neighboring anchors were calculated.

Although range based localization can achieve high accuracy, these algorithms are expensive in terms of the hardware and consumed power. This is not desirable for the wireless sensor nodes that are supposed to be low cost and generally operate on battery power. Therefore, several localization solutions have been proposed using range free algorithms. These algorithms aim at estimating the location of a target node without a need to estimate the pair-wise distances by means of ranging. One of the commonly used range free localization algorithm is DV-Hop [17]. This algorithm exploits the network topology for computing the position of sensor nodes. Specifically, it first determines the hop-distance (i.e., the number of hops) between pairs of anchor nodes by discovering the shortest paths between the two nodes. Since the location coordinates for anchor nodes are already known, the average distance per hop along these paths can be calculated by dividing the Cartesian distance between them by the corresponding number of hops. The sensor nodes, which know their hop-distance from anchors, can then use the average distance per hop for computing their estimated distance from every anchor in the network. Once the distances are estimated, triangulation can be used to estimate the location of the target node.

Due to the low positioning accuracy of the DV-Hop algorithm, researchers have proposed several improvements in the basic algorithm. For example, an improved twice-refinement DV-Hop localization algorithm that uses the RSSI auxiliary ranging and an error correction mechanism based on target sensor node’s neighborhood centroid is claimed to have improved the localization accuracy [18]. Another improved version of the DV-Hop algorithm, known as Hybrid DV-Hop algorithm [19], uses both the network topology and the RSSI values between the neighboring nodes, thus, resulting in a significant increase in the localizing accuracy.

Considerable research work has been reported on applying learning algorithms for localization in WSNs. For example, fuzzy logic-based approach [20], kernel-based learning [21], maximum likelihood parametric approaches [22], statistical regression [23], support vector machine (SVM) classification [24,25,26,27,28,29], and deep learning algorithms [30,31], all have been proposed from machine learning algorithms. The SVM algorithm probably dominates the reported work due to its strong learning ability, being robust against noise, and ability to generalize well. It has been used both as a classification algorithm [24,25,28,32,33] as well as regression algorithm [34]. In order to use it as a classification algorithm for localization, the deployed area is divided into cells such that each node resides in a cell. The algorithm first fits a model on the training examples (of anchors) with known classes. The classifier is then used for predicting the membership of non-anchor nodes to one such segment based on the decision trees. If a node is classified as $\left[c_{x_{i}},c_{y_{i}}\right]$ that means it is (assumed to be) located in the cell with those coordinates.

Some researchers used hop-count connectivity of non-anchor nodes to one or more anchor nodes, thus, relaxing the condition of one-hop connectivity for the non-anchor nodes [24,27]. In that case, the training set consists of ordered tuples generated by anchor nodes. A given training example generated by an anchor node, in turn, consists of the hop count distance between that anchor node and every other anchor node in the network and the location of that anchor becomes the label for the training example. Aimed at improving the prediction performance of this algorithm, a probabilistic SVM was proposed that modifies the SVM’s sign function [25]. The modified function applies the sigmoid function on the unbounded real number (that is the output of SVM sign function) to get a real number between in [0,1]. The estimation accuracy can further be improved by post-processing. Specifically, once all nodes have been localized, a technique known as mass spring optimization can be applied. The idea is to treat each node as mass and its link as springs. Every node needs to attain balance by moving to a position where the net force of springs is zero.

Also, since the SVM is a quadratic programming (QP) problem that has time and space complexities of $O(n^{3})$ and $O(n^{2})$, respectively, the size of training dataset might need be reduced in order to reduce the processing time, especially for larger wireless sensor networks. It is also known that the optimal hyperplane that determines the classification is constructed by the support vectors. Thus, other non-support vectors are less useful (because they have no impact of the optimal position of hyperplane) for training the SVM and thus can be removed [28]. Some researchers have used a similar approach for formulating the localizing problem using received the signal strength metric as the basis for machine learning algorithm in a similar way to us [24,35]. The work reported in there, however, is different from ours in terms of the feature set and subsequent machine learning model. For example, we define feature vectors that use signals from all nodes, including anchor as well as non-anchor nodes, in the network as opposed to just the signals from anchor nodes. That formulation of feature vectors is, in fact, close to one of our approaches, named a reduced feature vector, which we actually used in a different way from the reported work. However, if the signals only from the anchors are used, then there might be many sensor nodes in the network, which need to be localized, but would not receive signals from all (or most) of the anchor nodes. In such a case, one has to use a secondary metric, i.e., hop count, for localizing those nodes [35]. The use of hop counts as an input parameter to a localizing algorithm normally results in increased estimation errors. In our proposed algorithm, we don’t use hop-count at all and solely rely on the received signal strength values.

## 3. Machine Learning Algorithms for Localization

Machine learning algorithms can be divided into two broad categories, namely supervised algorithms and unsupervised algorithms. The former category, which is applicable to localization in wireless sensor networks, can further be broken down into regression algorithms and classification algorithms. Both of these approaches have been utilized for localization in the reported work.

Using a supervised machine learning algorithm consists of two phases, namely the training phase and prediction phase. In the first phase, training examples are used for the algorithm to learn the underlying correlations among the training instances. The training dataset for localization in WSN consists of the feature vectors associated with every anchor node (also known as predictor variables) and its true location coordinates (also known as predicted variables), which are already known. The feature vector of every anchor node, as explained in the following sub-section, consists of an n-tuple of RSSI values as measured by that anchor node for the signals received from other nodes. During the training phase, the learning algorithm essentially fits a statistical model on the training dataset. This is realized by determining the values of a set of parameters, defined by the model, as optimally as possible. The model is used in the next phase.

The second phase involves estimating the location coordinates of the sensor nodes by using the model defined in the last phase. A sensor node can be localized only if it has its own feature vector (i.e., the n-tuple) with the same composition as used for training data. This feature vector of the target wireless sensor nodes is used as input to the learned model and it produces (or predicts) the location coordinates as its output. As the location of the sensor nodes is estimated, the parametric values of the model remain unchanged. This implies the order of sensor nodes being localized has no bearing on the estimated location coordinates for these nodes. This provides more flexibility as compared to the case of traditional trilateration-based progressive localization algorithm approach.

Localization of wireless sensor nodes in a two-dimensional plane requires the estimation of two coordinates for the target nodes, which generally are real numbers. It is thus a regression problem rather than a classification one. The feature set that can be used for generating the training data includes the measurements of RSSI values at different locations in the deployed area while the output variables are the two coordinates that specify the location. Since the system has two output variables, a multi-output regression algorithm should ideally be used for estimating or predicting the location coordinates of the target nodes. However, we have not found any published work that has used multi-output regression model for network-wide localization in WSN. Thus two models, one for each coordinate, are independently fit on the training data.

While using a machine learning algorithm for localizing in wireless sensor networks, one faces several trades-off as listed below. These trade-offs not only can influence the choice of the model to be used but can also affect the accuracy of the estimated locations.


**Regression vs. Classification**
Support vector machine (SVM) have been dominantly proposed for localizing wireless sensor nodes. The SVM algorithm, however, is a classification algorithm and thus requires the localization problem be mapped into a classification problem. In order to realize that, a common approach is to divide the deployment area into (rectangular or square) cells and then each node is classified based on its membership to these cells. Even in the best assumed case of the nodes being classified with 100% accuracy, the estimation error is proportional to the size of the cell. For a large scale WSN, that might be a significant localizing error. However, this may be acceptable depending on the target application. In any case, one faces a trade-off. Specifically, a smaller cell size might offer better accuracy while a bigger cell size might offer better efficiency or faster localization.
**Size of Training Data Set**
Machine learning algorithms require sizeable training datasets for fitting a suitable model. A good model is the one that captures the features faster and then generalized over the test data very well. The problem of underfitting and overfitting in machine learning algorithms need to be handled carefully. A smaller feature set results in underfitting while a larger feature set requires larger training data in order to avoid overfitting. For a localization algorithm, the training data is normally generated by the anchor nodes. The number of these nodes, however, is assumed to be small in a WSN. This offers another trade-off that needs to be carefully handled in order to get a model that generalizes well over the sensor nodes (the ones that need to be localized) in the network.
**Multivariate vs. Univariate Modeling**
As mentioned before, localization of a node involves estimation of its Cartesian coordinates, i.e., two coordinates on a plane or three in space. Most of the proposed algorithms, however, use models which predict only one dependent variable (also called prediction) while using many independent (or predictor) variables. A relatively straightforward way to deal with this issue is to train two machines for independently estimating each coordinate. In such a case, the two independently fitted models fail to capture any correlation that existed in the prediction variables. Specifically, for the localization problem, the radio signals travel straight from the transmitting node to the receiving node. That distance has a certain relationship with the Cartesian coordinates (i.e., square root of the sum of squares of the two components). One has to deal with this issue. We have tried using multivariate (or multi-output) regression from Matlab but that proved only to be a wrapper for two independent models predicting a single variable each.

### Defining the Feature Vectors

One of the most crucial tasks in designing a machine learning based solution is devising an appropriate feature set for the system model that captures the underlying correlations between predictor and predicted variables well. Here we describe the rationale behind our formulation of the feature vectors for the wireless nodes and explain how these feature vectors capture the underlying correlations with the location of nodes. It is worth noting that every wireless node in the network receives radio signals from its immediate neighbors. These neighboring nodes might include both the anchor and non-anchor nodes. Let us consider an arbitrary ith node, an anchor or a non-anchor one, in the network. Assuming that all nodes in the network have similar radios and using the same transmission power, every signal received at the ith node will have a specific RSSI value that depends on the distance between the corresponding transmitter node and the ith node. If there are a total of *N* nodes in the network, we can define a row vector of length *N* such that the kth element, 1≤k≤N, in that vector is actually the RSSI value for the signal transmitted by the kth wireless node. If the kth node is physically located close to some ith node, the received signal will be strong whereas if the kth node is far away from it, the received signal will be very weak and the corresponding entry in the vector will have a negligibly small value. Since any two nodes in the network have their own separate physical location, their corresponding distances from all other nodes will be different and thus their vectors will be different too. If the two nodes are located very close to each other, their vectors will be dissimilar by a small value and that small dissimilarity corresponds to the small distance between them. In essence, these vectors vary as the location of nodes varies and the change in the vector is correlated with the direction of change in location. Our goal is to capture that correlation for the anchor nodes and then exploit it for estimating the location of non-anchor nodes.

## 4. Proposed Algorithms

This section describes the channel model used in the simulations, system model for our target system, i.e., network-wide localization, and the machine learning models that were used for applying machine learning algorithms.

### 4.1. Channel Model

Since we have used RSSI values in our simulations to define the feature vectors for our machine learning algorithms, it is prudent to describe the channel model used for generating these values. However, since the RSSI is a relative scaling of signal strength that varies across radio chipsets, we have used a more standardized and absolute measure of signal strength, i.e., the received signal power measured in dBm on a logarithmic scale. On this scale, a value closer to 0 dBm implies good quality received signal. In fact, we have used a well-known channel pathloss model, called the log-normal shadowing model, which consists of measuring the power present in a radio signal as received by a node across a distance and calculating the propagation loss.

Our channel model defines the received signal power as given below [1].
(1)$$RSS\left(d\right)\left(\mathrm{dBm}\right)=P_{t_{x}}-P_{loss}\left(d_{0}\right)-10\eta log_{10}\frac{d}{d_{0}}+X_{\sigma }$$where RSSI(d) denotes the signal power at the receiver node located across a distance *d* from the transmitting node, $d_{0}$ is the reference distance, $P_{t_{x}}$ denotes the power of transmitted signal in dBm, $P_{loss}\left(d_{0}\right)$ is the signal power loss across the reference distance $d_{0}$, η is the path loss exponent whose value depends on the medium of propagation, and $X_{\sigma }$ is the noise, which is modeled as a Gaussian random variable with zero mean and σ as its standard deviation.

### 4.2. System Model

In this section, a system model for the simulated wireless sensor network is defined. Let the network consist of *N* nodes including *m* anchor nodes and N−m non-anchor nodes. In a typical WSN, both the anchor and non-anchor nodes function are essentially similar physical wireless sensor nodes. While the anchor nodes know their own location, the non-anchor nodes need to be localized. For the sake of simplicity, we henceforth call the non-anchor nodes as sensor nodes. It is assumed that anchor nodes might not have all sensor nodes within their transmission range. Also, our system does not require that every sensor node must be residing with the transmission range of at least three anchor nodes. That might be required for the triangulation algorithm and in this part of our study we don’t use triangulation algorithms at all. In order to define the training data set, we note that all wireless nodes (anchors as well as the sensor nodes) can transmit and receive signals from their immediate neighbors. It is worth mentioning that a wireless node might be able to receive signals from only a subset of all nodes. We also assume that nodes are not changing their location during the localization phase.

In order to define the training data set, we note that all wireless nodes (anchors as well as the sensor nodes) can transmit/receive signals to/from their immediate neighbors. It is worth mentioning that a wireless node might be able to receive signals from only a subset of all nodes. We also assume that nodes are not changing their location during the localization phase. The data from m anchor nodes is used for training a machine learning algorithm. Let us define the vector $H_{aa_{i}}$ to be a tuple such that
(2)$$H_{aa_{i}}=<h\left(A_{i},A_{1}\right),h\left(A_{i},A_{2}\right),h\left(A_{i},A_{3}\right),...,h\left(A_{i},A_{m}\right)>,$$where $h(A_{i},A_{j})$ is the RSSI measurement for the signal as recorded by the anchor node $A_{i}$ from every anchor node $A_{j}$, 1≤i,j≤m, in the network. It is worth mentioning that if the node $A_{i}$ is located out of the transmission range of some node $A_{j}$, 1≤j≤m, the corresponding element $h(A_{i},A_{j})$ in the vector $H_{aa_{i}}$ will have a zero as its RSSI value. Also, the value for $h(A_{i},A_{i})$ in $H_{aa_{i}}$ will be set to some maximum value RSSImax. Let us now define another vector $H_{as_{i}}$ to be a tuple such that
$$H_{as_{i}}=<h\left(A_{i},S_{1}\right),h\left(A_{i},S_{2}\right),h\left(A_{i},S_{3}\right),...,h\left(A_{i},S_{N-m}\right)>,$$where $h(A_{i},S_{k})$ is the RSSI value recorded by the anchor node $A_{i}$, 1≤i≤m, for the signal transmitted every sensor node $S_{k}$, 1≤k≤N−m, in the network. Again, if the node $A_{i}$ is located out of the transmission range of some node $S_{j}$, 1≤j≤N−m, the corresponding element $h(A_{i},S_{j})$ in the vector $H_{as_{i}}$ will be set to the value of zero.

We can similarly define two more vectors $H_{sa_{i}}$ and $H_{ss_{i}}$ such that
$$H_{ss_{i}}=<h\left(S_{i},S_{1}\right),h\left(S_{i},S_{2}\right),h\left(S_{i},S_{3}\right),...,h\left(S_{i},S_{N-m}\right)>,$$and
$$H_{sa_{i}}=<h\left(S_{i},A_{1}\right),h\left(S_{i},A_{2}\right),h\left(S_{i},A_{3}\right),...,h\left(S_{i},A_{m}\right)>,$$where $h(S_{i},S_{j})$ and $h(S_{i},A_{k})$ are the RSSI measurements as recorded by the sensor node $S_{i}$ for the signals received from the sensor node $S_{j}$, 1≤j≤N−m and anchor node $A_{k}$, 1≤k≤m, respectively. Also, the value for all $h(S_{i},S_{i})$, 1≤i≤N−m, will be set to an appropriate maximum value RSSImax and for $h(S_{i},S_{j})$ and/or $h(S_{i},A_{k})$ to zero if the sensor node $S_{i}$ is located out of the transmission range of corresponding sensor node $S_{j}$ and anchor node $A_{k}$.

Given these vectors, we are now ready to define the feature vectors for use in our machine learning algorithms. Specifically, these feature vectors will be used for specifying the training data set and test data set. In fact, we define two different types of feature vectors, called as, the reduced feature vector and extended feature vector.


**Reduced Feature Vectors**


A reduced feature vector for a node (anchor or sensor) consists of *m* RSSI values corresponding to the signals received from m anchor nodes. Reduced feature vectors ignore the signals transmitted by the sensor nodes. So, the training data for a machine learning algorithm can be defined as the set of all ordered pairs ($H_{aa_{i}},y_{i}$), 1≤i≤m, where $H_{aa_{i}}$ is the RSSI vector, as defined in Equation (Equation 2), for anchor node $A_{i}$, 1≤i≤m, and $y_{i}$ is this node’s real position. The test data set, on the other hand, consists of vectors $H_{sa_{j}}$, 1≤j≤N−m for all sensor nodes in the network.


**Extended Feature Vectors**


As stated above, the reduced feature vectors ignore the signals transmitted by the sensor nodes. It seems, however, to be prudent to exploit these signals too for the network-wide localization in the network. There are several reasons for justifying that exploitation. Some noteworthy ones are listed below.

First of all, sensor nodes are a crucial asset in any wireless sensor network. The existence of these nodes must be exploited in every possible manner in order to maximize the utilization of network resources.Moreover, and rather more importantly, the signals transmitted by a sensor node and received by other (anchor and sensor) nodes have underlying correlations with their locations. These correlations need to be exploited for estimating their locations in the network.The number of anchor nodes in a WSN is relatively much smaller than that of sensor nodes. By including data from the non-anchor nodes, the training data would augmented to result in better localizing accuracy.Another subtle observation that we made during our studies is that systems with an extended feature vector are less sensitive to the degradation in channel link quality. We observed that strong links play more a dominant role in learning by the localization algorithms. On the other hand, weak links or missing links (i.e., nodes being out of transmission range) play a secondary role. As a result, reduced feature vector based systems suffered badly to the degradation in channel quality. On the other hand these systems responded much more positively to the incrementally increased transmission power of wireless nodes. That is further discussed in the next section.

An extended feature vector for a node (anchor or sensor) consists of *N* RSSI values corresponding to the signals received from m anchor nodes as well as N-m sensor (i.e., non-anchor) nodes. Let us define a vector $H_{train_{i}}$ for every anchor node $A_{i}$, 1≤i≤m, such that
$$H_{train_{i}}=\left[H_{aa_{i}}H_{as_{i}}\right]$$where the size of this vector is 1xN. The training data set can now be defined as the set of vectors $H_{train_{i}},$1≤i≤m, for all *m* anchor nodes and their corresponding location coordinates, i.e.,

(3)$$H_{train}=\left\{\left(H_{train_{1}},y_{1}\right),\left(H_{train_{2}},y_{2}\right),...,\left(H_{train_{m}},y_{m}\right)\right\}$$

It is worth noting that the data $H_{train}$ can be arranged in an m×(N+1) matrix.

The extended feature vector for the sensor (i.e., non-anchor) nodes can be defined in a similar fashion. Specifically, the sensor node $A_{j}$, 1≤j≤N−m, has its extended feature vector
$$H_{test_{j}}=\left[H_{sa_{j}}H_{ss_{j}}\right]$$such that $H_{test_{j}}$ is a vector of size 1xN. The test data set, $H_{t}est$, can then be defined as follows.

$$H_{test}=\left\{H_{test_{1}},H_{test_{2}},...,H_{test_{N-m}}\right\}$$

The test data can be arranged in an (N−m)×N matrix such that its ith row consists of the values from vector $H_{test_{i}}$, 1≤i≤N−m.

During this study, we aimed at studying the performance of both the linear regression as well as the support vector machine for localizing N-m sensor nodes. We developed two sets of simulations, one for each training data from reduced feature vectors and the extended feature vectors. These data sets were used to fit a model in each case. Once a learning algorithm gets trained, the estimated location of all sensor nodes $S_{k}$, 1≤k≤N−m, can be calculated by using the model learned during the training.

### 4.3. Regression Models

Numerous models have been defined for different machine learning algorithms. Some of the commonly used models include linear regression model, multiple linear regression model, neural networks, support vector machines with different types of kernels, Logistic classification or regression, *k*-nearest neighbors clustering, and random forest clustering. For the network-wide localization problem, we used two regression models, i.e., multiple regression model and SVM regression model with RBF kernel because treating localization problem as a clustering or classification problem is superficial in its nature. In this section, we briefly describe these two models before describing the simulation implementation and the results thus achieved.

#### 4.3.1. Multiple Regression Model

The linear regression is probably the simplest model used in machine learning algorithms with its own advantages. In fact, we have found it more promising for localization than more complex SVM based models.
$$h_{\theta }\left(x\right)=\theta_{0}x_{0}+\theta_{1}x_{1}+\theta_{2}x_{2}+...+\theta_{n}x_{n}$$or
(4)$$h_{\theta }\left(x\right)=\theta^{T}x$$where
$$\theta =\left(\begin{array}{c} \theta_{0}\\ \theta_{1}\\ .\\ .\\ .\\ \theta_{n} \end{array}\right)\mathrm{and}\;x=\left(\begin{array}{c} x_{0}\\ x_{1}\\ .\\ .\\ .\\ x_{n} \end{array}\right)$$

We now define a cost function $J(\theta_{0},\theta_{1},...,\theta_{n})$ such that
(5)$$J\left(\theta_{0},\theta_{1},...,\theta_{n}\right)=\frac{1}{2m}\sum_{1}^{m}{\left(h_{\theta }\left(x^{\left(i\right)}\right)-y^{\left(i\right)}\right)}^{2}$$where $x^{\left(i\right)}$ and $y^{\left(i\right)}$ be the features vector and the true location, respectively, of the ith training example (i.e., anchor node), 0≤i≤m, in the system. The goal for the machine learning algorithm is to fit a model with parameters $\theta_{0},\theta_{1},...,\theta_{n}$ while minimizing the cost function $J(\theta_{0},\theta_{1},...,\theta_{n})$ on to the training data set. In order to determine the model parameters, we used gradient descent algorithm that iteratively computes the gradient.

#### 4.3.2. Support Vector Machine Regression Model

One of the most popular machine learning algorithms is named support vector machines (SVM), which is mostly used for classification. In machine learning context, a support vector machine (SVM) is a supervised learning model, along with associated learning algorithm, used to analyze training data and then for classification and regression analysis. The SVM is also known as large margin classifier because it aims at defining a decision boundary with the largest possible margin between the instances of the two classes. It achieves that by mapping these instances in n-dimensional feature space onto an optimal (n−1)-dimensional hyperplane that provides the biggest margin between the two groups. The SVM can be used for solving linear and nonlinear classification problems. If the training data encompasses a nonlinear feature space, it will map the samples into a high dimensional feature space by using a nonlinear mapping function. The dimension of the new feature space might be large or infinite. However, the training examples become linearly separable in the new space to allow a model to fit well on these training examples. The SVM was later proposed to be used for regression [36]. Now most of the available packages such as Matlab, Octave, and Python Scikit offer multiple flavors of SVM regression.

The cost function for a binary SVM classifier is defined as given below.

(6)$$\min_{\theta }C\sum_{i=1}^{m}[y^{\left(i\right)}cost_{1}\left(\theta^{T}x^{\left(i\right)}\right)+\left(1-y^{\left(i\right)}cost_{0}\left(\theta^{T}x^{\left(i\right)}\right)\right]+\frac{1}{2}\sum_{i=1}^{n}\theta_{j}^{2}$$

This can be written as:$$f\left(w\right)=\frac{{\left||w|\right|}^{2}}{2}+C{\left(\sum_{i=1}^{m}\xi_{j}\right)}^{k}$$subject to the constraint
$$\xi_{i}=max(0,\left(1-y_{i}\left(wx_{i}+b\right)\right)$$where both *C* and *k* are user-specified parameters representing the penalty of incorrectly classifying the training instances.

During the training phase, as opposed to the case of linear and logistic regression algorithms where $\theta^{T}x\geq 0$ for y=1 and $\theta^{T}x<0$ for y=0, the SVM model ensures $\theta^{T}x\geq 1$ for y=1 and $\theta^{T}x<-1$ for y=0. That is why SVM is called wide margin classifier. Once the learner has been trained, predictions can be made by using the parameters of the fitted model. Specifically, we can predict y=1 if
$$\theta_{0}+\theta_{1}f_{1}+\theta_{2}f_{2}+...+\theta_{n}f_{n}\geq 0$$for an unlabeled input feature set. A y=0 is predicted otherwise.

The SVM model offers much better robustness by the use of kernel functions. A kernel is a similarity function that is used by a learning algorithm to quantize the pair-wise similarity between the training samples. This similarity function is then used to define a new powerful feature set, rather than the original feature set, for fitting a model on the training dataset. So, given *m* training examples
$$\left(x^{\left(1\right)},y^{\left(1\right)}\right),\left(x^{\left(2\right)},y^{\left(2\right)}\right),\left(x^{\left(3\right)},y^{\left(3\right)}\right),...,\left(x^{\left(m\right)},y^{\left(m\right)}\right),$$we compute the new feature set $f^{\left(i\right)}=<f_{1}^{\left(i\right)},f_{2}^{\left(i\right)},f_{3}^{\left(i\right)},...,f_{m}^{\left(i\right)}>$ for ith training example $\left(x^{\left(i\right)},y^{\left(i\right)}\right)$ as follows.
$$\begin{array}{cc} f_{1}^{\left(i\right)} & =similarity(x^{\left(i\right)},l^{\left(1\right)})\\ f_{2}^{\left(i\right)} & =similarity(x^{\left(i\right)},l^{\left(2\right)})\\ f_{3}^{\left(i\right)} & =similarity(x^{\left(i\right)},l^{\left(3\right)})\\ \;\vdots \\ f_{m}^{\left(i\right)} & =similarity(x^{\left(i\right)},l^{\left(m\right)}) \end{array}$$


It is worth mentioning that the objective function specified in (Equation 6) is minimized by using feature vector $f^{\left(i\right)}$ computed for each example $\left(x^{\left(i\right)},y^{\left(i\right)}\right)$. The kernel functions enable a linear learning algorithm to fit a non-linear model (or decision boundary) by transforming nonlinear spaces into linear ones. There are several types of commonly used kernels including, but not confined to, linear kernel, Gaussian kernel, radial basis function (RBF) kernel, polynomial kernel, etc. Since we are modeling a wireless communications system where the Gaussian based models are commonly used to characterize channel fading, we decided to use Gaussian kernel during our simulations for this project. The jth feature in higher dimensional space for the ith training example $\left(x^{\left(i\right)},y^{\left(i\right)}\right)$ can be calculated by using Gaussian kernel is specified in the following
(7)$$\begin{array}{cc} f_{j}^{\left(i\right)} & =similarity(x^{\left(i\right)},l^{\left(j\right)})\\ & =exp(-\frac{\left|\right|x^{\left(i\right)}-l^{\left(j\right)}{\left|\right|}^{2}}{2\sigma^{2}}) \end{array}$$where $l^{\left(j\right)}$ is the jth landmark and $x^{\left(i\right)}$ is the original feature vector consisting of *m* or *N* elements, depending on reduced or extended feature vector being used, for that input training example. It is worth mentioning that, in fact, the $x^{\left(i\right)}$ from ith training example is used as $l^{\left(i\right)}$.

## 5. Performance Analysis

In this section we analyze the performance of the proposed machine-learning localizing algorithms. We first describe a two-dimensional simulated system model and discuss how this model can be extended for solving localization problems in the higher dimensions. Finally we present the performance results followed by a discussion.

### 5.1. Simulated System Model

Our default simulated network consisted of 120 sensor nodes and 40 anchor nodes deployed in a 100 × 100 m${}^{2}$ field. The sensor nodes were deployed randomly during all simulations, thus, having a different layout in every simulation run. For the deployment of anchor nodes, we tested two scenarios. In the first case, anchor nodes were deployed randomly just like their non-anchor counterparts. The position of anchor nodes was thus changed for every run of the simulation. In the second scenario, anchors were deployed along a mesh grid in the deployment area. So the position of anchor nodes remained unchanged during all simulation runs for a given tested configuration. We have observed that the grid deployment of anchor nodes results in significant improvement in the localization accuracy. Also, for every tested configuration, all the proposed localizing algorithms were simulated using the same network layout in order to get better comparison results for their performance. During all these simulations, we have used the channel model as defined by Equation (Equation 1). Specifically, we have used a default value of 50 mW for $P_{tx}$, 2 for the path loss exponent η, and a Gaussian random variable with zero mean and σ=3 as standard deviation for the noise $X_{\sigma }$.

However, the default values for the above network parameters were suitably varied in specific experiments as shown in Table 1. For example, while studying the effect of anchor population on the localization accuracy, we varied the number of anchors from 20 to 60, with an increment of 5, while keeping all other system parameters unchanged at their default values, thus, making it to be nine different system configurations. Also, the results reported in this paper are based on a hundred simulation runs for every tested configuration.

It is worth mentioning that we have implemented our simulations for localization in a two-dimensional system just as an illustration for the sake of simplicity. The proposed models, however, can easily be extended for localizing nodes in three dimensions. It is worth noting that the system model defined by Equation (Equation 3) in Section 4.2 actually uses RSSI values for signals transmitted by anchor and non-anchor nodes. It is then fed into the two regression models specified by Equations (Equation 5) and (Equation 7) in Section 4.3. For a two-dimensional localization, each regressor is independently trained twice, once for each location coordinate, while using the same training dataset. So in case of localization of the target nodes in three dimensions, the regressor would independently be trained three times, corresponding to each of the three location coordinates.

### 5.2. Simulation Results

As stated before, we have defined two formulations, i.e the reduced features vector and extended features vector as described in the previous section, for mapping the localizing problem into a machine learning problem. We then used several machine learning regression models, including multiple regression and SVM regression using linear, RBF, and polynomial kernels, for simulating the system and predicting the location coordinates of the wireless sensor nodes. However, the performance results presented here are based on two particular regression models, i.e., multiple regression model and SVM using RBF kernel, because the performance results from the other models were only slightly different from the results presented here. The only target performance metric during these simulations was the localization accuracy under different system parameters as listed in Table 1. We present our findings by first plotting the localization accuracy, measured as the root mean square error (RMSE) as shown in Figure 1, against different test parameters of Table 1 for simulated networks where the anchor nodes were deployed as a grid in the deployment area. We then plot empirical cumulative distribution function (CDF) of the localization error for some of our test results, as shown in Figure 2, in order to get a deeper insight into localizing performance of the proposed algorithms. We also study two cases of deploying anchor nodes, i.e., deploying these nodes randomly versus deploying them as a grid and show our results in Figure 3. In addition, we present a set of snapshots, in Figure 4, depicting the final layout of the network in order to provide a post-localization view of the network. Finally, we have a technical discussion about the work reported in this paper.

(a)
**Sensor Node Population vs. Localization Error**
Our first experiment was aimed at studying the localization performance of a WSN as more wireless sensor nodes are added into it while keeping the number of anchor nodes unchanged. Since the models with reduced feature set use the training data generated by only anchor nodes, the localization accuracy should not change as the network size grows because the number of anchor nodes remains constant. That is indeed the case as shown in Figure 1a. On the other hand, the regression models using the extended feature set perform differently because these models use the signals from the sensor nodes too for training the learning algorithm. So the additional connectivity data generated by the added sensor nodes results in better training of the learning algorithm. The multiple regression model, therefore, slightly improves its localization accuracy as the network size grows. The localizing performance of the SVM regressor with RBF kernel, however, slowly degrades as more sensor nodes are added. It seems to be suffering from over-fitting because the length of feature vector (which depends on the total number of nodes) increases while that of training examples (which is equal to number of anchor nodes) remains constant.(b)
**Anchor Node Population vs. Localization Error**
Another aspect that might be of interest while deploying a self-localizing WSN is how the localizing performance improves when more anchor nodes are added into the network. The graph in Figure 1b depicts the results of our experiments in this regard. An interesting observation is that the models using the reduced feature set get the most benefit as more anchors are gradually added into the network. For example, the reduced feature SVM model reduces the localizing error by almost 50% in that graph. That is not unexpected because the reduced feature sets are more sensitive to the anchor population in the network. More anchors generate a larger, and probably better, training dataset that captures and exploits the underlying correlations in a better way. The extended feature models, on the other hand, improve their localization accuracy less significantly as more anchor nodes are added to the network. Since anchor nodes are normally more costly (in terms of material and deployment costs), that result can be exploited to reduce the deployment cost by using only the minimum number of anchors. Deploying too many anchor nodes has insignificant improvement in the localizing performance while using extended feature models.(c)
**Wireless Signal Quality vs. Localization Error**
Figure 1c depicts how the degradation in operating environment (i.e., wireless signal quality) affects the localization accuracy in the network. The system parameter σ specifies how much fading a signal is experiencing. To recall, $X_{\sigma }$ in Equation (Equation 1) is the shadow fading random variable which follows zero mean Gaussian distribution with σ as the standard deviation. Higher the value of σ, greater the fading effect on received signals. In this set of simulations, we varied the value of σ from 1 to 10 while keeping the number of anchor and sensor nodes at the default values of 40 and 120, respectively. It can be noticed in the graph that the localization error significantly increases when the value of σ is gradually increased especially beyond the value of 4. The SVM regression model with extended feature set experiences significantly more degradation in the localization accuracy. The SVM regression model with reduced feature set, on the other hand, shows much better response to degraded signal quality. That difference in behavior of the two SVM regression models is not unexpected due to the fact that the RBF kernel is exponential in its nature as is evident from Equation (Equation 7). The SVM regressor using extended feature set results in too many features, based on lower quality received signals, in its feature vector and the exponential nature of the kernel magnifies the embedded error in the recorded RSSI values. The SVM regressor using reduced feature set results in a feature vector which is far smaller in size as compared to the extended feature vector. So it does not suffer from overfitting and thus behaves much more gracefully as the value of σ is gradually increased. The multiple regression model with reduced feature set suffers the most because, unlike the SVM regressor, it does not generate higher degree feature vector. Rather it uses a feature vector that consists of actually RSSI values, which fails to capture the underlying correlations while using low quality signals. That results in a significant increase in the localization error. The multiple regression model with extended feature set, on the other hand, endures the degradation in signal quality well (i.e., localization error increases almost linearly) because it exploits the signals from the sensor nodes too.(d)
**Radio Transmission Power vs. Localization Error**
In this experiment, the transmission power was gradually increased from 10 mW to 100 mW while keeping other parameters unchanged. It can easily be observed in Figure 1d that no significant gains in localization accuracy were recorded as the transmission power of the wireless nodes was increased. Increasing the transmission power, on the other hand, has serious implications on the network’s operating life. Since a wireless sensor node’s effective life is as long as its battery life, it would be prudent to set the transmission power of wireless nodes just high enough for reliable communications that would result in the increased functional life of those nodes. (e)
**Another View: Plotting Cumulative Distribution Function**
It is also useful to analyze the system performance by using cumulative distribution function of a performance metric for having a better insight of the system behavior. We have used it for some of our simulation results for better understanding the performance of proposed localizing algorithms. Specifically, the graphs in Figure 2a,b plot the CDF of localization error for two scenarios of anchor population tests using the two extended features regression models, the multiple regression model and SVM regression model. For these tests, all input parameters, except the number of anchors, were set to their respective default values as shown in Table 1. The number of anchor nodes used during test is 20 and 55, respectively. These nodes were deployed as a grid in the deployment area. It can be observed in these graphs that multiple regression model was able to localize about 70% of sensor nodes with a localizing error of up to 5 meters by deploying 20 anchor nodes and more than 90% sensor nodes when using 55 anchor nodes. The corresponding values for the SVM model stand at approximately 51% and 72%, respectively. Further, some of the sensor nodes in both cases have large localization error that is a cause of concern and should be the focus of our future research effort.Similarly the next two graphs in Figure 2 show how the localization accuracy rapidly drops when the overall wireless signal quality deteriorates by changing the value of σ from 1 to 6. It can be observed in the graphs that multiple regression model was able to localize 85% sensor nodes with an accuracy of up to 5 meters in Figure 2c and only 20% sensor nodes in Figure 2d. The SVM regression model suffers even more as the value of σ is changed from 1 to 6. Again some of the sensor nodes have a very high localization error. (f)
**Anchor Nodes: Random vs. Grid Deployment**
An important aspect of launching a wireless sensor network, especially a large one, is the placement of anchor nodes. While placing these nodes at randomly chosen locations might be easier, their placement in a grid might be beneficial in terms of improved localization accuracy. We have studied this aspect too in our simulations and the results are shown in Figure 3. In these graphs, we plot the RMSE for two extended feature models, i.e., SVM with RBF kernel and multiple regression model, for both cases of deploying anchor nodes randomly as well as along a grid. During these simulations, we have used the same system parameters as specified in Table 1. Also, the same number of simulations runs (i.e., 100) for every test configuration was used during these experiments. It can be seen in these graphs that the deploying anchor nodes in a grid significantly improves the localization accuracy as compared to the case when these nodes are deployed at random locations. For example, in Figure 3a, the random deployment results in an average of 18.1% and 10.4% increase in the localizing error over the grid deployment for the SVM regression model and multiple regression model, respectively. For the graph in Figure 3b, the increase in the error stands at 34.6% and 23.5%, respectively, for the two deployment scenarios. The corresponding numbers for the graph in Figure 3d are 18.3% and 11.2%. However, the deployment of anchor nodes in a grid probably has the greatest effect on the overall connectivity in the network because anchor nodes are uniformly spread in the deployment area. That should result in a much better localization accuracy especially when the transmission power of wireless nodes is set to a lower level.(g)
**Overall Localizing Performance**
While it is important to analyze the localizing performance of a localizing algorithm by varying different system parameters and then plotting the results (as we have done above), it is useful to look at the overall final picture to get a sense of how the proposed algorithms performed and how the performance can be improved. Figure 4 shows a set typical localizing scenarios depicting the actual positions of 120 sensor nodes, marked by blue circles, and the corresponding estimated node positions, marked by blue stars while the solid lines between the pairs show the displacement or error in localization. The figure also shows the positions of 40 anchor nodes marked as red squares. In all these scenarios, the learning algorithms used only the extended feature sets. Figure 4a,b show the nodes localized by using multiple regression model and SVR regression model, respectively with anchor nodes randomly deployed in the network. The same random layout for sensor nodes was used in both cases. It is easily noticeable that both machine learning algorithms were able to localize sensor nodes towards the center of the deployment area much better than the nodes closer to the boundary. Moreover, the multiple regression model outperformed the SVR model while localizing the nodes closer to the boundary because the displacement arcs are fewer and smaller in Figure 4a than in Figure 4b. A similar pattern can easily be seen in Figure 4c,d where the anchor nodes were deployed along a grid. Again both deployments mutually share the same random layout for the sensor nodes. The multiple regression model once again performs much better, especially for nodes closer to edges of the deployment area, than the SVM regression model with RBF kernel. That is a bit surprising because the RBF kernel is Gaussian in its nature that should be able to fit a model onto the RSS values from the log-normal fading channel much better than by the multiple regression model, which is essentially a multi-variate linear model. It requires extended time and effort to study and determine the cause of that anomaly and that could be a subject of another research paper.

## 6. Discussion

While applying machine learning models, one has to determine a balance between the number of features and that of training examples in order to avoid overfitting or underfitting. An overfitted model normally does not generalize very well on the test examples. Increasing the number of training examples, in such a case, normally helps to alleviate the problem of overfitting. A smaller feature set, on the other hand, may result in underfitting. Increasing the number of training examples, in this case, does not really help but increasing the number of features, especially with higher degree, results in much better learned model.

The extended feature set results in overfitting, especially for the SVM regression with RBF kernel, because there are relatively too many features in the feature set as compared to the number of training examples. Specifically, the size of the feature vector for every training example is *N*, which is total number of nodes in the network, while the size of the training dataset is *m*, which is the number anchor nodes in the network. Since in a typical wireless sensor network, the number of anchor nodes *m* is much smaller than the number of sensor nodes, the number of features in every training example is much larger than the number of training examples. One way to reduce the impact of that imbalance is to reduce the number of features in the feature vector. In our system, however, that is not feasible for two reasons. First, the feature set is generated from the RSSI values for the signals received from all one hop neighboring wireless nodes. The set of these neighboring nodes varies across the network. So it is not possible to remove the RSSI values for signals transmitted by some specific nodes from the training set because the removed nodes might have been the only neighboring anchors for many sensor nodes. Second, the actual feature set is, in fact, generated by the chosen kernel from the vectors containing RSSI values. Modifying the kernel code is unsafe and out of the scope of this research effort.

It needs to be mentioned that we have not implemented a classification based machine learning localizing algorithm for performance comparison with our proposed algorithms due to certain reasons. First, classification based localization requires substantially more computational resources, especially for larger deployment areas. As an example, if we break down a 100 m × 100 m deployment area into 10 m × 10 m cells, the number of resulting classes is twenty. Using one vs. all strategy for multi-class classification, one would need to train twenty learning machines. Smaller cell size would result even greater number of classes. That is substantially more demanding in terms of computational resources. Also, it is currently not clear how to compare the results from classification and regression based localizing algorithms. Specifically, classification based localization offers a trade-off between accuracy versus efficiency. Smaller cell size offers better accuracy but greater number of classes. Regression based localization algorithms offer no such trade-off. So, it’s not obvious which cell size should be selected for performance comparison of a classification based localizing algorithm with a regression based localizing algorithm. These and other issues forced us to leave this aspect for future investigations.

## 7. Conclusions

In this paper, we have presented the results of our investigation on using machine learning algorithms for network-wide localization in large-scale wireless sensor networks. Contrary to the approach of treating the localization as a classification problem, as reported in the most of the published work, we treat it as a regression problem. Starting with describing different factors and trade-offs that must be considered while using machine learning algorithms for localization, we defined a set of novel feature vectors in order to capture the underlying correlation between the wireless signals and the location of wireless sensor nodes. We then used different machine learning models, specifically the multiple regression model and SVM regression model with RBF kernel, for training the learning algorithms with the training data set. We have studied the localizing performance of these models while varying network parameters such as number of anchor and sensor nodes, transmission power, and the link quality. We have also studied the impact of deploying the anchor nodes in a grid rather then at randomly chosen locations in the deployment area. Based on the performance results, we then presented an analysis to get some insights in the behavior of these learning algorithms. The results from this research effort will make a basis for our future work that would aim at improving the learning models through ensemble approaches and refinements in the feature set. Another important direction that we would like to pursue is investigating the ways to compare the performance of classification based localizing algorithms with that of regression based algorithms and their respective suitability to different deployment scenarios and applications.

## Figures and Tables

**Figure 1 sensors-18-04179-f001:**
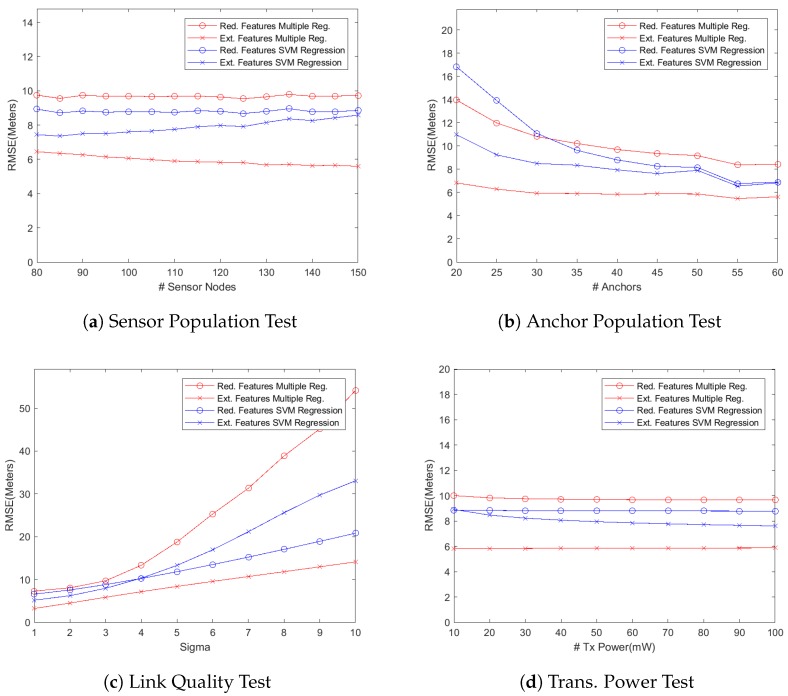
Network with anchors deployed as a grid: Impact of varying different system parameters on the performance metric (i.e., localization accuracy).

**Figure 2 sensors-18-04179-f002:**
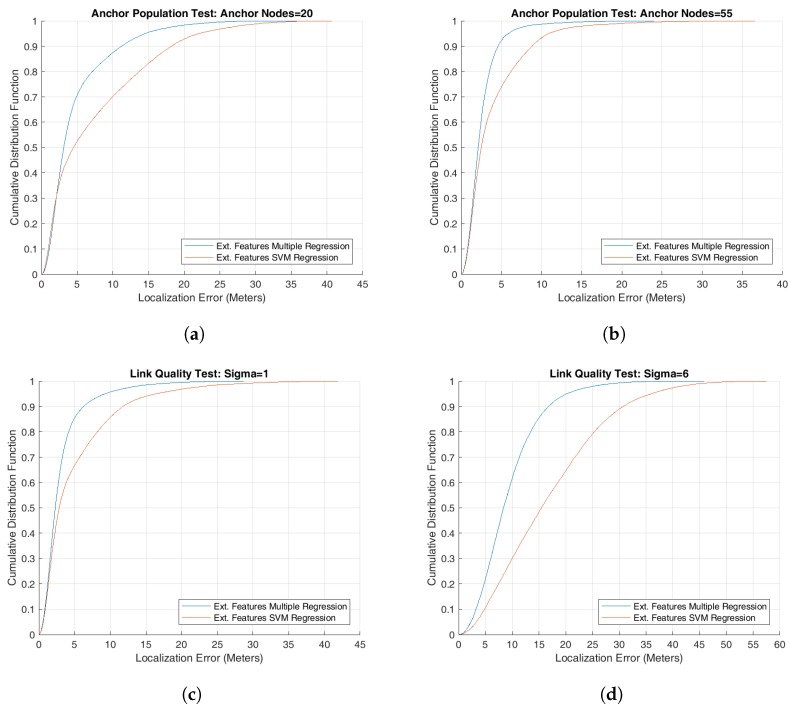
Empirical cumulative distribution function of localization error plotted for the extended feature set regression models for anchor population and link quality tests.

**Figure 3 sensors-18-04179-f003:**
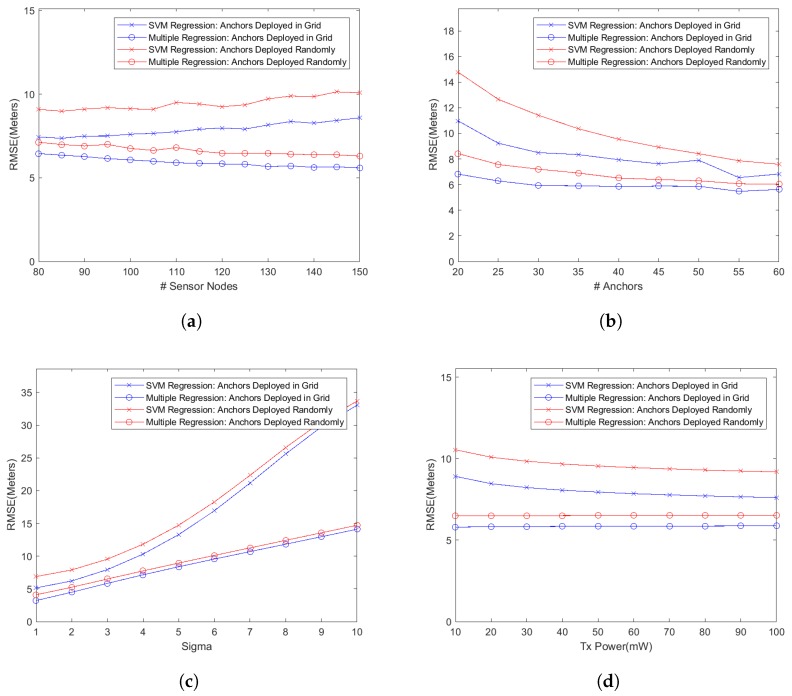
Impact of deploying anchor nodes randomly versus along a grid on the performance metric (i.e., localization accuracy) for extended feature set regression models.

**Figure 4 sensors-18-04179-f004:**
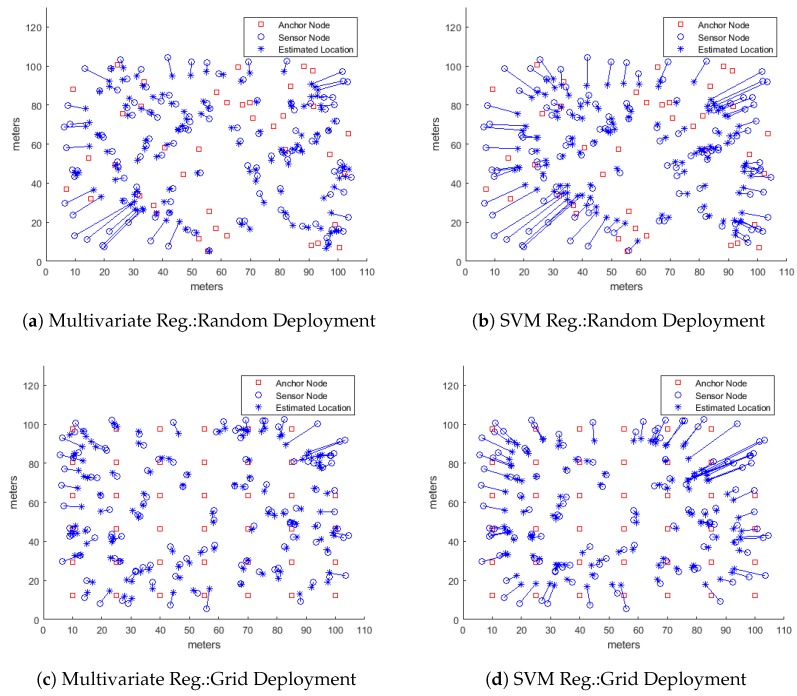
Final layout showing localized nodes in the network using machine learning algorithms when anchor nodes are deployed randomly and in a grid.

**Table 1 sensors-18-04179-t001:** Simulated system parameters used in different experiments.

Test Type	Num. Anchor Nodes	Num. Sensor Nodes	Tx Power (mW)	Signal Quality (Sigma)
Sensor Population Test	40	80–150	50	3
Anchor Population Test	20–60	120	50	3
Tx Power Test	40	120	10–100	3
Signal Quality Test	40	120	50	1–10

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
