# Peer review of "Machine Learning Based Localization in Large-Scale Wireless Sensor Networks"

_sensors, 2018, doi:10.3390/s18124179_

Round 1
Reviewer 1 Report
In the paper localization of nodes in wireless sensor network based on machine learning algorithms is proposed and implemented in Matlab and tested using simulations.
The theoretical part of the paper is well written, however, the results section could be improved. I would suggest to provide more information on characteristics of localization error. It would be nice to see cumulative distribution function of localization errors, to better understand performance of tested algorithms. From the RMSE it is hard to understand what the distribution of error is.
I also suggest to run more simulations, in the paper it is stated "we have run at least three simulations" . It is not quite clear what does that mean, did you run 3 simulations, or 4 simulations or 1000 simulations?
The higher the number of simulations, the better. Moreover, it is not clear how were anchor nodes placed in the area, placement of anchor nodes might have huge impact on localization accuracy. I would suggest to run more simulations (with random placement of anchor nodes) and run some simulations with fixed positions of anchor nodes in the same shape (placement in grid maybe?).
Author Response
Dear Sir,
First of all I would like to really thank you for a very valuable feedback. Please see the point-wise response to your comments/feedback.
The theoretical part of the paper is well written, however, the results section could be improved.
Thanks. So I have done a major revision of the results part of the paper in order to accommodate reviewers' comments and suggest to my best.
It would be nice to see cumulative distribution function of localization errors, to better understand performance of tested algorithms. From the RMSE it is hard to understand what the distribution of error is.
I have updated the description of the results to make it easier to understand. I have also more graphs depicting the CDF of the error for better understanding of the error distribution.
I also suggest to run more simulations
Now the reported results are based on a hundred simulation runs for every tested configuration. I have first tried 20, 50, and 80 simulation runs for each configuration. But the curves are smoother for 100 runs per configuration. I hope that addresses your concern.
Moreover, it is not clear how were anchor nodes placed in the area, placement of anchor nodes might have huge impact on localization accuracy.
That is a very nice point. I never thought of that before. The deployment pattern (or a lack of that) of anchor nodes indeed has a significant impact on the localizing error in the network. I have experimented with both the random deployment and the deployment along a grid and the results have been plotted as CDF graphs. That is a very useful addition to the results section in the paper. Thanks for pointing that aspect out to me.
I’m giving the final touches to the draft and hope to upload it in a day or two.
Best Regards,
Dr. Ghulam Bhatti
Taif University
KSA
Reviewer 2 Report
The paper presents the investigation results of using machine learning techniques for network-wide localization in wireless sensor networks (WSNs). The localization problem is mapped into machine learning models, and it is treated as a regression problem. The multivariate regression model and the support vector machine regression model are exploited in the paper.
The paper is well written and easy to follow. The machine learning techniques are effectively applied to the localization problem in WSNs.
It is suggested that the simulation parameters should be summarized in a table for readers in Section 5. In addition, the studied performance metrics should be clearly defined.
No comparison with the conventional localization schemes (e.g., classification algorithm) is given in the paper. In order to claim the superiority of the presented scheme over the conventional approaches, the comparison should be given in addition in Section 6.
Author Response
Dear Sir,
Thanks a lot for giving a very useful feedback after reviewing the manuscript. I have tried my best to update the draft in the light of your and other reviewers’ comments/feedback.
The paper is well written and easy to follow. The machine learning techniques are effectively applied to the localization problem in WSNs.
Thanks.
It is suggested that the simulation parameters should be summarized in a table for readers in Section 5. In addition, the studied performance metrics should be clearly defined.
A table showing all the configuration parameters and their default/test values has been added to the results section. I agree that it makes understanding the experiments and the results much easier. The studied performance metric has also been adequately elaborated.
No comparison with the conventional localization schemes (e.g., classification algorithm) is given in the paper. In order to claim the superiority of the presented scheme over the conventional approaches, the comparison should be given in addition in Section 6.
First of all, we are not really claiming that our regression based algorithms are better than those based on classification (even though that might indeed be true). Rather we are claiming that our formulation of the problem (i.e. defining the features vectors and then mapping them onto the regression models) is unique and more natural.
Having said that, I would really like to have a performance comparison of the classification based localizing algorithms with my regression based algorithms. However, it is a subject of a new manuscript by itself. As an example, if we break down a 100m x 100m deployment area into 10m x 10m cells and then classify each location coordinate as one of 10 classes along x-axis and same along y-axis, we need to train 20 learning machines (using one vs. all strategy) for every simulation run. And I have 100 simulation runs for every tested configuration reported in this manuscript. That would need a lot of computing resources and time. Now let just roughly check the best possible accuracy of classification based approach. So if all nodes are localized correctly (i.e. which cell every node is residing in) by the algorithm, we still have an average error of more than 7 meters (because a node can be located anywhere in the cell). If the algorithm fails predicting with 100% accuracy (which is more probable than not) then localizing error will be even bigger. So I think I can write a paper investigating these and related aspects of those algorithms.
I’m currently giving the final touches to the draft and hope to upload it in a day or two.
Best Regards,
Dr. Ghulam Bhatti
Taif University
KSA
Reviewer 3 Report
This paper presents some results concerning localization performed using a machine learning approach. The paper is well organized and generally easy to read, eve if some improvements are required, as suggested in the following.
The parameters which appear in formula (1) (\eta, P_loss, X_\sigma) are crucial when performing indoor localization in real scenario. Authors should explain which parameters are used to obtain simulation results shown at the end of the paper.
Authors focus on 2D localization but, it general, localization is performed in 3D environments. Authors should justify their choice and explain whether the proposed approach is applicable also in 3D scenarios and how it could be extended to take into account the third dimension.
It would be interesting to see whether these simulation results are in agreement with experimental ones, i.e., results obtained in a real environment.
Authors state that they run at least three simulations for each scenario. The number three is apparently low and the number of simulations should be increased, or at least better motivated.
Structure of the paper:
Sec 5. is very short. Please merge it with Sec. 6.
English concerns:
Some sentences should be rewritten as they are not clear, e.g.:
- Sentence starting at line 143
- Sentence starting at line 194
English check is required, e.g.:
- line 250: We have aimed
- 4.3.1 Simplist
In general, please avoid to write short sentences followed by a sentence starting with “But”. This makes the paper more difficult to be read.
Author Response
Dear Sir,
First of all, thanks a lot for giving a very useful feedback after reviewing the manuscript. I have tried my best to update the draft in the light of your and other reviewers’ comments/feedback. Please see the response to your comments in the following.
This paper presents some results concerning localization performed using a machine learning approach. The paper is well organized and generally easy to read, even if some improvements are required, as suggested in the following.
Thanks a lot. Really appreciated.
The parameters which appear in formula (1) (\eta, P_loss, X_\sigma) are crucial when performing indoor localization in real scenario. Authors should explain which parameters are used to obtain simulation results shown at the end of the paper.
We have described the used values for different parameters of this equation in the simulation model.
A table showing all the configuration parameters and their default/test values has also been added to the results section. That makes understanding the experiments and the results much easier. The studied performance metric has also been adequately elaborated.
Authors focus on 2D localization but, it general, localization is performed in 3D environments. Authors should justify their choice and explain whether the proposed approach is applicable also in 3D scenarios and how it could be extended to take into account the third dimension.
We have justified the use of 2D localization and described how the model can easily be extended for localization in 3D environments. Appropriate changes have been made in the simulations section.
It would be interesting to see whether these simulation results are in agreement with experimental ones, i.e., results obtained in a real environment.
That would be real nice. But we are talking about getting data from larger-scale wireless sensor networks. That is beyond our resources and out of scope for the research project that the paper has been produced for. I hope we’ll be able to get data from some external sources in future to compare our current results and then further expand our research effort.
Authors state that they run at least three simulations for each scenario. The number three is apparently low and the number of simulations should be increased, or at least better motivated.
Now the reported results are based on a hundred simulation runs for every tested configuration. I have first tried 20, 50, and 80 simulation runs for each configuration. But the curves are smoother for 100 runs per configuration. I hope that addresses your concern.
The deployment pattern (or a lack of that) of anchor nodes has a significant impact on the localizing error in the network. We have also experimented with both random deployment and the deployment along a grid and the results have been plotted as CDF graphs. That is a very useful addition to the results section in the paper.
I’m currently giving the final touches to the draft and hope to upload it in a day or two.
Best Regards,
Dr. Ghulam Bhatti
Taif University
KS
Round 2
Reviewer 1 Report
"In the first case, the anchor nodes were deployed randomly just like their non-anchor counterparts."
Does that mean that position of anchor nodes was changed in each run of simulation?
409: "scenaro" should be "scenario" , "the" is not used in plural ("the anchors nodes" is incorrect)
411, 421, 442: "localizing" should be replaced by "localization"
It is not clear why path loss exponent was set to 1, typical values are between 2 and 4, 2 being used for free space (1 would give even lower attenuation).
Would it be a problem to have even more than 100 simulation runs (maybe just for the case you decide to analyze further using CDF)?
Are results presented in Figure 1 based on both random and mesh grid placements of anchor nodes? It is not clear from the manuscript.
Figure 1 (c) shows that reduced features can perform better than extended features for RBF regression when higher noise level is present in channel. This is, however, in contrast with description on page 14, line 490: " However, the effect is more pronounced on the two reduced feature set models."
Is SVM in the figure 2 the same as RBF in figure 1? (If yes, same names should be used thru the paper.)
Was extended or reduced feature set used to achieve results presented in figure 2?
Presented CDF plots are somehow questionable, it is not clear under which conditions the results were achieved.
It should be analysis of error under fixed conditions. Seems figures show something else. Details on conditions under which the results were achieved have to be provided.
Moreover, I would suggest to use "cdfplot" function in Matlab to plot CDF of localization error.
Can you provide layout of localized nodes for mesh grid?
It maybe included in the figure 3 with different marks and color so we will have direct comparison.
Author Response
Thanks again for your valuable feedback. Please see my responses in the following. Some of your comments are not clear to me. I'll really appreciate if you could please elaborate a little more on those points.
Regards,
Dr. Ghulam Bhatti
------------------------------------------------------------------------
"In the first case, the anchor nodes were deployed randomly just like their non-anchor counterparts."
Does that mean that position of anchor nodes was changed in each run of simulation?
Yes, that is indeed the case. I have added the following sentence in the manuscript. “The position of anchor nodes was changed for every run of the simulation.”
409: "scenaro" should be "scenario" , "the" is not used in plural ("the anchors nodes" is incorrect)
Fixed.
411, 421, 442: "localizing" should be replaced by "localization"
All instances of “localizing accuracy” replaced with “localization accuracy”. Thanks pointing that out. I really feel happy by learning something new.
It is not clear why path loss exponent was set to 1, typical values are between 2 and 4, 2 being used for free space (1 would give even lower attenuation).
It was my mistake. It was actually set to 2 (for free space) in the simulation code. But I have now changed it to 3 for sub-urban area.
Would it be a problem to have even more than 100 simulation runs (maybe just for the case you decide to analyze further using CDF)?
No problem. It’s only more time consuming. Since I have run the simulations again with new value of path lost exponent, I’ll run simulations 200 times per tested configuration.
Are results presented in Figure 1 based on both random and mesh grid placements of anchor nodes? It is not clear from the manuscript.
Results presented in Figure 1 are based on mesh grid placements of anchor nodes.
It was stated in the manuscript on line 445: “…shown in Figure 1, for a simulated network having anchor nodes deployed as a grid.”
Figure 1 (c) shows that reduced features can perform better than extended features for RBF regression when higher noise level is present in channel. This is, however, in contrast with description on page 14, line 490: "However, the effect is more pronounced on the two reduced feature set models."
My characterization is wrong, I think. Higher values of sigma result in lower quality of received signals for all transmissions (from anchor and non-ancho nodes alike). So quality of the training data from anchor nodes (reduced feature set) will be degraded in the same way as that of the data from all nodes (extended feature set). I’ll have to revise the description for case (C) on page 13.
Is SVM in the figure 2 the same as RBF in figure 1? (If yes, same names should be used thru the paper.)
Both refer to the same algorithm. I have fixed by changing the name in Figure 1.
Was extended or reduced feature set used to achieve results presented in figure 2?
Extended feature set was used for getting results reported in Figure 2. That was stated in the manuscript, line 512: “graphs, we plot the cumulative distribution function of the localizain error for the two extended feature models, i.e. SVM with RBF kernel and multiple regression model, for both cases of deploying anchor nodes randomly as well as along a grid.” I’ll also mention this in the caption of Figure 2.
Presented CDF plots are somehow questionable, it is not clear under which conditions the results were achieved. It should be analysis of error under fixed conditions. Seems figures show something else. Details on conditions under which the results were achieved have to be provided. Moreover, I would suggest to use "cdfplot" function in Matlab to plot CDF of localization error.
I don’t understand what you mean “it is not clear under which conditions the results were achieved.” In fact, two curves (for the cases when anchors were deployed in a grid) in each graph, we used the same results as shown in Figure 1 (where we plotted RMSE against the varying input parameter) but we decided to plot only the results for extended feature models (otherwise there would have been too many curves in each plot). The simulations for both the gridded and random cases were run under the same test configurations (conditions?). I don’t know what you want me to add in the description.
Can you provide layout of localized nodes for mesh grid? It maybe included in the figure 3 with different marks and color so we will have direct comparison.
Sure. I’ll make Figure 3 have 4 parts. Two new sub-figures will show the final layout of the network. I assume you want to see the placement of anchor nodes. We don’t show anchors in part (a) and part (b) because it becomes too congested. But I’ll show anchors in the gridded layouts.
Reviewer 2 Report
The revised version of the paper has been responded to the review comments and revised in part.
However, the performance study is not satisfactory in terms of comparison over the conventional approaches in the revised version. At least the authors should state (a) why the performance comparison is not given in the paper and (b) it is their future work as responded to the review comment, in the conclusion section.
Author Response
Dear Sir,
Thanks for your prompt review of the revised manuscript. I really appreciate your time and effort. Please see my response to your comments.
However, the performance study is not satisfactory in terms of comparison over the conventional approaches in the revised version. At least the authors should state (a) why the performance comparison is not given in the paper and (b) it is their future work as responded to the review comment, in the conclusion section. I have added the following paragraph on page 16, line 565 (under ‘Discussion’)
“It needs to be mentioned that we have not implemented a classification based machine learning localizing algorithm for performance comparison with our proposed algorithms due to certain reasons. First of all, classification based localization requires substantially more computational resources, especially for larger deployment areas. As an example, if we break down a $ 100m $ x $ 100m $ deployment area into $ 10m $ x $ 10m $ cells, the number of resulting classes is twenty. Using one vs. all strategy for multi-class classification, one would need to train twenty learning machines. Smaller cell size would result even greater number of classes. That is substantially more demanding in terms of computational resources. Also, it is currently not clear how to compare the results from classification and regression based localizing algorithms. Specifically, classification based localization offers a trade-off between accuracy vs. efficiency. Smaller cell size offers better accuracy but greater number of classes. Regression based localization algorithms offer no such trade-off. So, it’s not obvious which cell size should be selected for performance comparison of a classification based localizing algorithm with a regression based localizing algorithm. These and other issues forced us to leave this aspect for future investigations.”
I have also added the following lines at the end of Conclusions section
“Another important direction that we would like to pursue is investigating the ways to compare the performance of classification based localizing algorithms with that of regression based algorithms and their respective suitability to different deployment scenarios and applications.”
Regards,
Dr. Ghulam Bhatti
Round 3
Reviewer 1 Report
Currently, CDF plots consist of only 10 values. I assume same mean values that are presented in Figure 1, therefore, these CDF plots are wrong and their information value is 0, even worse they might be misleading.
Other than that the paper seems to be fine.
I do not think reviewer should give lecture on statistical analysis of data sets in his review, but well...
You should plot CDF from "raw" error values you get from simulations. That means you take vector of N errors (from N simulation runs) and plot it. That helps to understand other statistical parameters of error.
If you calculate errors for more than 1 "blind node" in each simulation run then total number of errors in vector will be M*N (M is number of localized nodes in each simulation run, N is number of simulations).
All simulations used to plot the CDF curve (single one) have to run under same set conditions, i.e. all parameters are fixed. So you have to have fixed SNR, number of anchor nodes, total number of nodes, transmit power...
You can include more CDF plots with different set of parameters if you need to prove something, or if there is some significant difference in distribution of errors. For example when number of anchor nodes is changed (other parameters should be set to same values as before to see impact of one variable).
With 200 simulations (and one localized node) the CDF figures should be relatively smooth.
It is not important to provide CDF plots for all variations of parameters you used in your simulations (just pick one or more as example).
Author Response
You should plot CDF from "raw" error values you get from simulations. That means you take vector of N errors (from N simulation runs) and plot it. That helps to understand other statistical parameters of error.
Thanks for your feedback. I wish you could have said that earlier. Anyhow, due to the pressure from the mid-term exams and ABET visit, I could not get it clearly what you really wanted. Anyhow, I hope it now has the required changes. I am sure the manuscript is in much better shape now. Thanks again.
Ghulam Bhatti